# Incidence and Predictors of Loss to Follow Up among Patients Living with HIV under Dolutegravir in Bunia, Democratic Republic of Congo: A Prospective Cohort Study

**DOI:** 10.3390/ijerph19084631

**Published:** 2022-04-12

**Authors:** Roger T. Buju, Pierre Z. Akilimali, Erick N. Kamangu, Gauthier K. Mesia, Jean Marie N. Kayembe, Hippolyte N. Situakibanza

**Affiliations:** 1Department of Public Health, Faculté de Medicine, University of Bunia, Bunia P.O. Box 292, Democratic Republic of the Congo; rogerbujutsedha1@gmail.com; 2Department of Biostatistics and Epidemiology, Kinshasa School of Public Health, University of Kinshasa, Kinshasa P.O. Box 11850, Democratic Republic of the Congo; 3Département des Sciences de Base, School of Medicine, University of Kinshasa, Kinshasa P.O. Box 11850, Democratic Republic of the Congo; erick.kamangu@unikin.ac.cd; 4Unité de Pharmacologie Clinique, School of Medicine, University of Kinshasa, Kinshasa P.O. Box 11850, Democratic Republic of the Congo; mesia.kahunu@unikin.ac.cd; 5Department Internal Medicine, School of Medicine, University of Kinshasa, Kinshasa P.O. Box 11850, Democratic Republic of the Congo; jm.kayembe@unikin.ac.cd (J.M.N.K.); situakibanza.nani@unikin.ac.cd (H.N.S.); 6Department of Tropical Medicine, Infectious and Parasitic Diseases, School of Medicine, University of Kinshasa, Kinshasa P.O. Box 11850, Democratic Republic of the Congo

**Keywords:** armed conflict, anti-retroviral therapy, HIV patient loss to follow up, Bunia, RDC

## Abstract

This study aimed to examine the incidence and predictors of loss to follow up (LTFU) in the context of ongoing atrocities caused by armed conflict, where HIV treatment programs and HIV-infected patients may face unique challenges in terms of ART adherence and retention in care. We conducted an observational prospective cohort study of 468 patients living with HIV (PLWHIV) under dolutegravir (DTG) in all health facilities in Bunia between July 2019 and July 2021. Kaplan–Meier plots were used to determine the probability of LTFU as a function of time as inclusive of the cohort. The main outcome variable was LTFU, defined as not taking an ART refill for a period of 3 months or longer from the last attendance for refill, and not yet classified as ‘dead’ or ‘transferred-out.’ The log-rank test was used to compare survival curves based on predictors. Cox proportional hazard modeling was used to measure predictors of LTFU from the baseline until 31 July 2021 (the endpoint). A total of 3435.22 person-months (p-m) were involved in follow up, with an overall incidence rate of 33.48 LTFU per 1000 p-m. Patients who had less experience with ART at enrolment and the ethnically Sudanese, had a higher hazard of being LTFU compared to their reference groups. This study reports a high LTFU rate in this conflict setting. An ART program in such a setting should pay more attention to naive patients and other particularly vulnerable patients such as Sudanese during the pre-ART phase. The study implies the implementation of innovative strategies to address this high risk of being LTFU, reducing either the cost or the distance to the health facility.

## 1. Introduction

Sub-Saharan Africa (SSA) remains the region most heavily affected by the AIDS epidemic, with two-thirds of the 38 million people infected with HIV in 2020 living there [1,2]. In the Democratic Republic of Congo (DRC), the national response to HIV/AIDS infection and STIs in the health sector has been greatly improved with the integration of the comprehensive HIV prevention, care and treatment package in 79% of the health zones in the DRC, including the province of Ituri [3].

The retention of people living with HIV (PLWHIV) on antiretroviral therapy (ART) at the various health care facilities remains a major challenge in these situations [4]. The province of Ituri continues to have a large number of unstable populations, including PLWHIV, mainly due to the exactions of armed conflict [5]. Previous studies have shown that the absence or cessation of ARV treatment can lead to a deterioration in clinical condition, a drop in immunity and an increase in viral load in patients who are off treatment. This contributes to an increase in morbidity and mortality related to HIV infection and to the development of resistance to antiretroviral drugs (ARVs) [6,7].

Studies of cohorts of PLWHIV on ART in SSA report dropouts, the rate of which varies according to the context. For example, in a cohort study of patients on ART in Kenya, approximately one-third of patients were either lost to follow up or reported dead within two years, with more than half of the attrition occurring within six months of starting ART [8]. A study was conducted in another city in the DRC on the impact of the disclosure of HIV status to family members on those lost to follow up [9].

This study is particularly important because it examines the situation of those lost to follow up (LTFU) in the context of the ongoing atrocities caused by armed conflict, where HIV-treatment programs and HIV-infected patients may face unique challenges in terms of ART adherence and retention in care.

## 2. Methods

### 2.1. Study Design and Participants

Between July 2019 and July 2021, we conducted an observational prospective cohort study of 468 patients living with HIV under DTG in all the health facilities in Bunia. Bunia is a city located in the eastern part of DRC. The city has seen armed conflict since 2017, and there are still areas where the conflict continues to this day. We included in this study patients who were 18 years or older (published in detail elsewhere) [10]. Pregnant women were excluded. All clinics are located in the city of Bunia. And the city of Bunia is surrounded by armed conflict zones where the troops of the regular army and the militias clash daily. Clashes continue from time to time in the city of Bunia. 

### 2.2. Procedures, Data Collection and Outcome

Upon inclusion, participants were switched to DTG at 50 mg plus lamivudine at 300 mg and tenofovir disoproxil fumarate at 300 mg, all taken orally once daily. Thereafter, scheduled visits for viral load were performed at six-month intervals. At all the visits, the participants underwent medical evaluation, including physical examination, the reporting of adverse events, a review of concomitant medications, and HIV-1 RNA viral load, hemogram and liver, urine and renal function tests [10]. ART refill appointments are made monthly in the health facilities. Patients’ CD4 cell and full blood counts (including hemoglobin) were scheduled every 3 months as part of the routine follow up.

A questionnaire was designed to gather sociological and ethnical data from the participants. Trained interviewers from the research team conducted the data collection. After pre-testing the first 10 patients, the questions were checked for consistency and modified as necessary. Designated supervisors monitored the data collection to ensure the validity and consistency of the data, as well as compliance with ethical guidelines (published in detail elsewhere) [10].

This study considered LTFU when a patient is not seen at the ART clinic for at least 90 days or 3 months after the last missed appointment, but not transferred out from the facility to another facility. A special procedure was in place for tracing patients’ LTFU. There was an ‘extended follow up’ in the health facilities. The ‘extended follow up’ involved home visits or phone calls using community health agents. Tracing was conducted by telephone contact with patients or their families using information collected at the health facility enrollment and home visits. The community health agents had completed high school education and received extra training on HIV/AIDS. They reported the status of each patient to the data clerks of each health facility after each visit. Each patient received 5 follow-up calls after three home visits from the community health agents. We defined the patient status after extended follow up as ‘Died’, if a family member, neighbor or community leader reported the death of the patient.

The following information was collected: socio-demographic characteristics (age, gender, marital status, education and residence), clinical characteristics (the status of treatment before enrollment, tobacco consumption, WHO HIV clinical stage, and exposure time under DTG-based regimen in months) and biological characteristics (serum creatinine at baseline). The status of treatment before enrollment was defined according to three categories: experienced patients, self-reported naïve patients with baseline VLs of <50 copies/mL and self-reported naïve patients with baseline VLs of ≥50 copies/mL [10].

The main outcome variable was LTFU, defined as not taking an ART refill for a period of 3 months or longer from the last attendance for refill, and not yet classified as ‘dead’ or ‘transferred-out.’ The date of the LTFU was defined as that of the last visit to the clinic according to the medical records.

### 2.3. Statistical Analysis

Data were recorded using the software Epi Info 7 Version 7.2.0.1. (Centers for Disease control and Prevention, Atlanta, GA, USA).The data were checked for completeness before entry. A pre-developed Epi Info-based data entry template was designed and given to the data entry clerks. Double entry of the data and thorough cleaning were also performed to ensure high quality.

Proportions were presented, and the main outcome variable was LTFU, for which the chi-square test or Fisher’s exact test was used when appropriate. The incidence rate for recorded LTFU events per 1000 person-months (p-m) was determined from the date of enrolment. For patients known to have been transferred out or deceased, data were censored at the date of the last appointment or death. Data on patients still in active care at the end of the study period were censored at the date of their last visit to the clinic. Kaplan–Meier curves were used to determine the probability of LTFU as a function of time of inclusion in the cohort. The log-rank test was used to compare survival curves based on determinants. Cox proportional-hazard modeling was used to measure predictors of LTFU from treatment induction to the endpoint, set at 31 July 2021. The following variables were included in the Cox regression model: gender, age, education, marital status, ethnic group, stage of disease, alcohol use, tobacco history, and status of treatment before enrolment. The proportionality test based on Schoenfeld residuals verified compliance with the assumption of proportionality of risks. Multicollinearity was assessed using variance inflation factors (VIFs) greater than 4.0 and was found to be insignificant—the mean VIF was 1.55. For the first paper from this study, all the tests were two-sided and the level of significance was set to *p* < 0.05. All the tests were performed using the Stata software version 14 (StataCorp, College Station, TX, USA). (Published in detail elsewhere [10]). 

### 2.4. Ethical Statement

The study protocol was approved by the institutional review board ethics committee for research subjects at the Kinshasa University School of Public Health (no. app.: ESP/CE/094/2018 of 9 August 2018). All the participants provided written informed consent prior to participation in the study. However, patients’ records/information were anonymized and de-identified prior to the analysis as previously published [10].

## 3. Results

### 3.1. Patient Characteristics at Enrollment and Follow-Up Status

A total of 468 patients living with HIV were included in this study. The background characteristics of the participants are reported elsewhere [10]. At baseline patients had a mean haemoglobin of 13.84 (SD = 2.77)

Overall, the patients LTFU were similar to those who were still in program, in the distribution analysis, in terms of age, gender, education, ethnic groups, baseline hemoglobin, and alcohol and tobacco use. However, the two groups were different in terms of the WHO stage of disease and status of treatment before enrolment. More LTFU patients had advanced disease (Stage III and IV) (61% vs. 50%; *p* = 0.038) and were less experienced on ART (48% vs. 61%; *p* < 0.001) (Table 1).

### 3.2. LTFU Rate

In this cohort, more than one-fourth (28, 8%; 95% CI: 24.9–33.1) were LTFU during the study period. A total of 3435.22 person-months (p-m) were involved in follow up, with an overall incidence rate of 5.82 deaths per 1000 p-m and 33.48 LTFU per 1000 p-m. As shown in the previous paper, 12.0%, 21.4%, 26.5%, 28.6% and 28.8% of patients were lost to follow up after 1, 3, 6, 9 and 12 months, respectively [10].

### 3.3. Predictors of LTFU among HIV-Infected Patients on ART

Patients who had less experience on ART at enrolment and the ethnically Sudanese (AHR = 2.03; 95% CI: 1.16–3.52) had a higher risk of being LTFU compared to their reference groups (Figure 1 and Figure 2). Patients who were in advanced disease (Stages III and IV) had a higher risk of being LTFU compared to those who were at Stage I or II (AHR = 1.40; 95% CI: 0.97–2.03); this relationship was not significant (*p* = 0.072) (see Table 2).

## 4. Discussion

To our knowledge, this is one of the few studies to quantitatively document the loss to follow up (LTFU) in a (post-) conflict setting in Sub-Saharan Africa [9]. We found that 29% of the HIV-infected patients in this study were LTFU. Patients who had less experience on ART at enrolment and the ethnically Sudanese had a higher risk of being LTFU compared to their reference groups.

The LTFU rate reported in this study is higher than that in the study findings reported previously [9,11,12,13,14,15]. The very high LTFU rate that we report in this cohort could be directly related to insecurity but also to the lack of social support. These two factors could be the main reasons for non-attendance at health facilities in a conflict setting such as Bunia. Indeed, the armed conflict in and around the city of Bunia leads to the displacement of many patients from home to their health facility, with consequences in terms of cost and insecurity risk in accessing their health facility. People living with HIV/AIDS (PLWHIV) who need social support lack it and may be tempted to abandon their treatment. Previous studies have reported that the travel time to health facilities and the associated opportunity cost (in terms of the financial cost or time that could be allocated to other things) are important barriers to the retention of patients in the ART program [13,16,17]. The long distance to the health facility and the risk of clashes or ambushes by militia and other armed groups in the city of Bunia could discourage patients from continuing treatment. Thus, many patients may decide to interrupt their antiretroviral treatment. Since the discontinuation of ART care and treatment threaten the effectiveness of ART [18], this implies the need to implement innovative strategies to address the high risk of being LTFU, reducing either the cost or the distance to the health facility. In Mozambique, the cost of travel was significantly reduced by patients living in the same area creating organized groups that took turns visiting health facilities and collecting drugs for all the group members [19]. A club of patients grouped according to the proximity of addresses, where a patient from the club would be chosen by his or her peers to collect the drugs for the group in order to reduce the risk of insecurity and travel costs when the situation did not allow all the patients to collect the drugs from the clinic.

In this study, naïve patients had a higher risk of LTFU compared to patients who had previous experiences with ART. The increased treatment experience in the health facility among experienced patients could explain their lower rate of LTFU than naïve patients. The naïve patients seem not yet to be used to the context of insecurity and the staff of the treatment center.

Our findings suggest that Sudanese patients had a higher risk of LTFU. A previous paper in the same setting showed that the Sudanese ethnic group had the poorest response under ART in Bunia [10]. The Sudanese seem to be a marginalized and food-insecure people compared to other ethnic groups in the city of Bunia [20,21]. Previous papers have shown that food insecurity is associated with poor adherence to treatment [22,23,24].

This study has several limitations, the main one being that the possibility of misclassification might be due to the sensitive nature of the disclosure of treatment status; individuals may incorrectly report on having disclosed their HIV status to others. The research team could not ascertain the true outcomes of the patients who were documented as being LTFU due to insecurity in the region over that period of time. We cannot exclude the potential confounding effect of some variables on LTFU, which were not collected in this study. We did not investigate whether some missed variables like the body mass index, status of mental illness, stigma and discrimination, distance to the nearest health facility, cell phone possession, and having a caregiver can influence LTFU.

Nevertheless, this study has the advantage of being among the few to document LTFU in a context of conflict and post-conflict. These results might help researchers, healthcare workers and other stakeholders involved in HIV treatment and care to understand the incidence and predictors of LTFU in conflict-affected settings.

## 5. Conclusions

This study reports a high LTFU rate in this conflict setting. An ART program in such a setting should pay more attention to naive patients and other particularly vulnerable patients, such as Sudanese patients, during the pre-ART phase. More targeted counseling and follow up is needed. The study implies that the implementation of innovative strategies to address this high risk of being LTFU, reducing either the cost or the distance to the health facility, is warranted. Further studies to analyze whether food security, status of mental illness, stigma and discrimination, distance to the nearest health facility, cell phone possession, and having a caregiver can influence LTFU.

## Figures and Tables

**Figure 1 ijerph-19-04631-f001:**
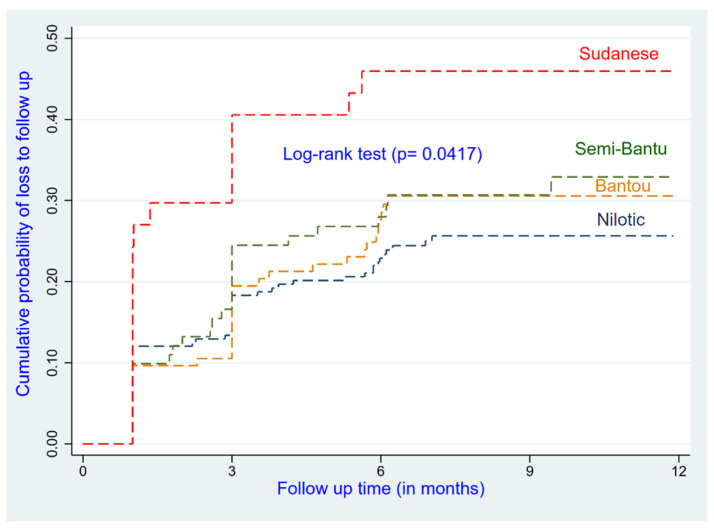
Cumulative incidence of LTFU by ethnic groups.

**Figure 2 ijerph-19-04631-f002:**
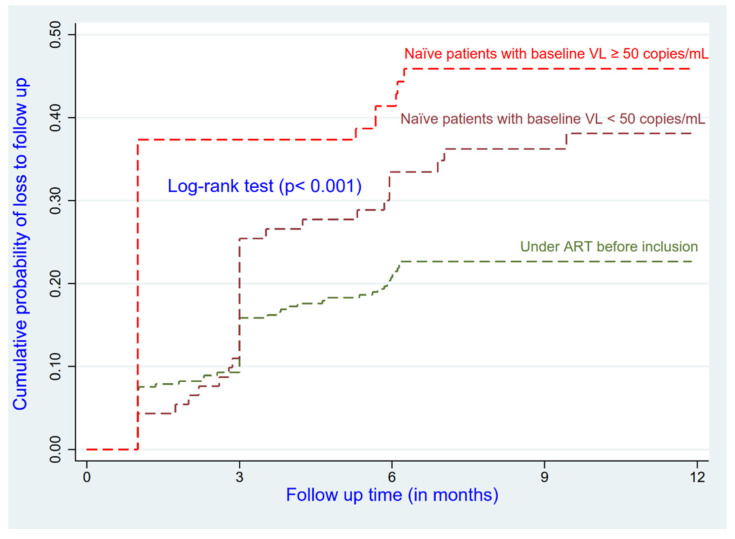
Cumulative incidence of LTFU by status of treatment at baseline.

**Table 1 ijerph-19-04631-t001:** Patient distribution of included first-time mothers and those lost to follow up by baseline characteristics, Bunia.

Background Characteristics	Total		LTFU		Still in Program		*p*-Value ^(1)^
	*n*	%	*n*	%	*n*	%	
Age							0.724
18–24	45	9.6	14	10.4	31	9.3	
25 and +	423	90.4	121	89.6	302	90.7	
Gender							0.698
Female	325	69.4	92	68.1	233	70.0	
Male	143	30.6	43	31.9	100	30.0	
Education							0.460
None/primary	307	65.6	92	68.1	215	64.6	
Secondary/tertiary	161	34.4	43	31.9	118	35.4	
Ethnic group							0.072
Nilotique	224	47.9	56	41.5	168	50.5	
Bantu	114	24.4	34	25.2	80	24.0	
Semi-Bantu	93	19.9	28	20.7	65	19.5	
Sudanese	37	7.9	17	12.6	20	6.0	
Alcohol							0.615
No	265	56.6	74	54.8	191	57.4	
Yes	203	43.4	61	45.2	142	42.6	
Tobacco use							0.340
No	357	76.3	99	73.3	258	77.5	
Yes	111	23.7	36	26.7	75	22.5	
WHO stage							0.038
Stage I and II	219	46.8	53	39.3	166	49.8	
Stage III and IV	249	53.2	82	60.7	167	50.2	
Status of treatment before enrolment							<0.001
Under ART	291	62.2	65	48.1	226	67.9	
New but with VL suppressed	93	19.9	33	24.4	60	18.0	
New but with high VL	84	17.9	37	27.4	47	14.1	
Hemoglobin, mean (standard deviation)	13.84 (2.77)		13.65 (2.91)		13.92 (2.72)		0.338
Total *	468	100.0	135	28.8	333	71.2	

*: percentage calculated among 468 participants. ^(1)^: from chi-square.

**Table 2 ijerph-19-04631-t002:** Multivariate analysis of predictors of LTFU.

	*n*	Events (LTFU)	Person-Months	Incidence Rate of LTFU	Adjusted HR	95% CI	*p*-Value ^(1)^
Age							
18–24	45	14	308.99	45.31	1.15	0.65–2.05	0.633
25 and +	423	121	3126.23	38.70	1		
Sex							
Female	325	92	2428.58	37.88	1		
Male	143	43	1006.64	42.72	0.90	0.60–1.35	0.597
Education							
None/primary	307	92	2137.25	43.05	0.92	0.62–1.36	0.675
Secondary/tertiary	161	43	1297.97	33.13	1		
Marital status							
Living alone	263	67	1994.38	33.59	1		
In union	205	68	1440.84	47.19	1.38	0.95–1.99	0.090
Ethnic group							
Nilotic	224	56	1675.37	33.43	1		
Bantu	114	34	877.95	38.73	1.19	0.76–1.86	0.458
Semi-Bantu	93	28	644.12	43.47	1.27	0.80–2.01	0.310
Sudanese	37	17	237.78	71.49	2.03	1.16–3.52	0.013
Alcohol							
No	265	74	1916.52	38.61	1		
Yes	203	61	1518.70	40.17	0.82	0.55–1.23	0.335
Tobacco use							
No	357	99	2568.82	38.54	1		
Yes	111	36	866.4	41.55	1.38	0.85–2.25	0.190
WHO stage							
Stage I and II	219	53	1795	29.53	1		
Stage III and IV	249	82	1640.22	49.99	1.40	0.97–2.03	0.072
Status of treatment before enrollment							
Under ART	291	65	2288.08	28.41	1		
New but with VL suppressed	93	33	676.68	48.77	1.62	1.06–2.48	0.026
New but with high VL	84	37	470.46	78.65	2.20	1.43–3.38	<0.001
Total	468	135	3435.22	39.30			

HR: Hazard ratio. ^(1)^: from Cox regression.

## Data Availability

Data are available upon request to pierre.akilimali@unikin.ac.cd.

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
