# Peer review of "Incidence and Predictors of Loss to Follow Up among Patients Living with HIV under Dolutegravir in Bunia, Democratic Republic of Congo: A Prospective Cohort Study"

_ijerph, 2022, doi:10.3390/ijerph19084631_

Round 1
Reviewer 1 Report
- What was the purpose behind the switch to DTG? (was it another study? Or guidelines based?)
- More breakdown on age should be given
- Do we have baseline CD4 counts and VL to include in the data? HgB?
- More description on the extended follow up: what was the length of time given? How many phone calls? Was it consistent across each patient?
- Map with geographic location of the clinics and the ethnic groups may help reinforce as well as areas of conflict to highlight the authors points
Author Response
What was the purpose behind the switch to DTG? (was it another study? Or guidelines based?)
Response: This is because of the National guidelines from July 2019 as stated in the previous publication: Buju RT, Akilimali PZ, Kamangu EN, Mesia GK, Kayembe JMN, Situakibanza HN. Predictors of Viral Non-Suppression among Patients Living with HIV under Dolutegravir in Bunia, Democratic Republic of Congo: A Prospective Cohort Study. Int J Environ Res Public Health. 2022;19(3):1085. Published 2022 Jan 19. doi:10.3390/ijerph19031085
- More breakdown on age should be given
This study included 468 patients living with HIV with a mean age of 38.97 (SD = 11.94) as stated in the previous publication: Buju RT, Akilimali PZ, Kamangu EN, Mesia GK, Kayembe JMN, Situakibanza HN. Predictors of Viral Non-Suppression among Patients Living with HIV under Dolutegravir in Bunia, Democratic Republic of Congo: A Prospective Cohort Study. Int J Environ Res Public Health. 2022;19(3):1085. Published 2022 Jan 19. doi:10.3390/ijerph19031085
- Do we have baseline CD4 counts and VL to include in the data? HgB?
We have baseline VL which was analysed in depth in previous publication: Buju RT, Akilimali PZ, Kamangu EN, Mesia GK, Kayembe JMN, Situakibanza HN. Predictors of Viral Non-Suppression among Patients Living with HIV under Dolutegravir in Bunia, Democratic Republic of Congo: A Prospective Cohort Study. Int J Environ Res Public Health. 2022;19(3):1085. Published 2022 Jan 19. doi:10.3390/ijerph19031085
Unfortunately, we did not collect baseline CD4.
At baseline patients had a mean haemoglobin of 13.84 (SD = 2.77), the patients LTFU were similar to those who were still in program, in the distribution analysis, of baseline haemoglobin.
- More description on the extended follow up: what was the length of time given? How many phone calls? Was it consistent across each patient?
This study considered LTFU when a patient is not seen at the ART clinic for at least 90 days or 3 months after the last missed appointment, but not transferred out from the facility to another facility. Each patient received 5 follow-up calls after three home visits from the community health agents.
- Map with geographic location of the clinics and the ethnic groups may help reinforce as well as areas of conflict to highlight the authors points
All clinics are located in the city of Bunia. And the city of Bunia is surrounded by armed conflict zones where the troops of the national army and the militias clash daily. Clashes continue from time to time in the city of Bunia. This is stated in the current version, however, we do not have an accurate map to show Bunia city and the clinics.
Reviewer 2 Report
This study reflects the conditions of PLWHIV, who are residents of conflict zones, which prevent them from having access to ART. The endpoint variable was LTFU, and the authors reported 29% LTFU. The authors found ethnic differences with increased risk in Sudanese.
The authors found that 29% of the HIV-infected patients in this study were LTFU. It is expected that patients with LTFU had a higher viral load.
The authors declare that the study follows from the recently published: Buju, R.T.; Akilimali, P.Z.; Kamangu, E.N.; Mesia, G.K.; Kayembe, J.M.N.; Situakibanza, H.N. Predictors of Viral Non-Suppression among Patients Living with HIV under Dolutegravir in Bunia, Democratic Republic of Congo: A Prospective Cohort Study. Int. J. Environ. Res. Public Health 2022, 19, 1085. https://doi.org/10.3390/ijerph19031085
The authors point out its limitations, but the study is especially relevant because it refers to social inequality due to armed conflict.
Authors should add if there were pregnant women in this Doultegravir follow-up cohort or they were excluded.
Author Response
This study reflects the conditions of PLWHIV, who are residents of conflict zones, which prevent them from having access to ART. The endpoint variable was LTFU, and the authors reported 29% LTFU. The authors found ethnic differences with increased risk in Sudanese.
The authors found that 29% of the HIV-infected patients in this study were LTFU. It is expected that patients with LTFU had a higher viral load.
The authors declare that the study follows from the recently published: Buju, R.T.; Akilimali, P.Z.; Kamangu, E.N.; Mesia, G.K.; Kayembe, J.M.N.; Situakibanza, H.N. Predictors of Viral Non-Suppression among Patients Living with HIV under Dolutegravir in Bunia, Democratic Republic of Congo: A Prospective Cohort Study. Int. J. Environ. Res. Public Health 2022, 19, 1085. https://doi.org/10.3390/ijerph19031085
The authors point out its limitations, but the study is especially relevant because it refers to social inequality due to armed conflict.
Responses: Thanks for the kind words
Authors should add if there were pregnant women in this Doultegravir follow-up cohort or they were excluded.
Response: Pregnant women in this Dolutégravir follow-up cohort were excluded.

Reviewer 3 Report
Dear authors
I’ve carefully revised your manuscript “Incidence and Predictors of Loss to Follow Up among Patients Living with HIV under Dolutegravir in Bunia, Democratic Republic of Congo: A Prospective Cohort Study”, this manuscript aims to examine the incidence and predictors of loss to follow up (LTFU) in the context of armed conflicts, where HIV treatment programs and HIV-infected patients may face unique challenges in terms of ART adherence and retention in care.
Here my comments to improve your work:
MAJOR
- Please revise your abstract for effectiveness and remember your abstract should be a brief summary of the manuscript. Abstracts should provide information in the Introduction, Methods, Results, and Discussion/Conclusions (IMRAD) order. And the abstract must stand on its own.
- Please make sure this is the correct approach to estimate incidence of new cases, and please explain how the rates vary among different subgroups or with different exposures?
- Please make sure to differentiate between sex and gender, please use sex when reporting biological factors and gender when reporting identity, psychosocial, or cultural factors.
- Please use more recent data for your epidemics, UNAIDS publishes HIV estimates every year.
- Please make sure to include more information about DRC, some data is available and I don’t see the need to use data from SSA.
- Please make sure to clearly state your inclusion and exclusion criteria.
- Please clarify if patients needed to meet certain criteria for DTG transition.
- Please consider revising your discussion and how you’re interpreting and comparing your results, considering the limitations and the strengths of your study, as you mentioned your country still continues to have areas with armed conflict.
- Please also include the strengths of your study and revise your limitations, they are generally more related with the design.
- Please make sure to discuss what are the implications of your research in actual clinical practice.
Minor comments
- Please revise the sub-headings of your table, not clear to me what do you want to say with overall (do you mean all participants, I will advise you to use instead “total”)
- Please revise your manuscript and the references for errors.
- Please consider using % as numbers instead of “one-fourth”, I will suggest you take your English readable internationally.
- Please use lower case n, as it denotes the number of people in a sample. An uppercase N represents the number of people in a given population.
Author Response
Dear authors
I’ve carefully revised your manuscript “Incidence and Predictors of Loss to Follow Up among Patients Living with HIV under Dolutegravir in Bunia, Democratic Republic of Congo: A Prospective Cohort Study”, this manuscript aims to examine the incidence and predictors of loss to follow up (LTFU) in the context of armed conflicts, where HIV treatment programs and HIV-infected patients may face unique challenges in terms of ART adherence and retention in care.
Here my comments to improve your work:
MAJOR
- Please revise your abstract for effectiveness and remember your abstract should be a brief summary of the manuscript. Abstracts should provide information in the Introduction, Methods, Results, and Discussion/Conclusions (IMRAD) order. And the abstract must stand on its own.
Response: We have provided information about the Introduction, Methods, Results, and Discussion/Conclusions according to the Journal requirement and there is word limit constraint
- Please make sure this is the correct approach to estimate incidence of new cases, and please explain how the rates vary among different subgroups or with different exposures?
Response: Incidence Rate of LTFU and the reported p-value is showing clearly where there is difference or not among different subgroups or with different exposures (see table 2)
- Please make sure to differentiate between sex and gender, please use sex when reporting biological factors and gender when reporting identity, psychosocial, or cultural factors.
Response: Thanks, We made this change
- Please use more recent data for your epidemics, UNAIDS publishes HIV estimates every year.
Response: We have used the most recent publications.
Mahy MI, Sabin KM, Feizzadeh A, Wanyeki I. Progress towards 2020 global HIV impact and treatment targets. J Int AIDS Soc. 2021;24 Suppl 5(Suppl 5):e25779. doi:10.1002/jia2.25779
- Please make sure to include more information about DRC, some data is available and I don’t see the need to use data from SSA.
Responses: We have available data we have and compare with data from other SSA countries.
- Please make sure to clearly state your inclusion and exclusion criteria.
Response: We made this edits
We included in this study patients who were 18 years or older (published in detail elsewhere) [10]. Pregnant women were excluded.
- Please clarify if patients needed to meet certain criteria for DTG transition.
Response: Authors did not decide switching process but the swich was decided since 2019 by the national guidelines
- Please consider revising your discussion and how you’re interpreting and comparing your results, considering the limitations and the strengths of your study, as you mentioned your country still continues to have areas with armed conflict.
- Please also include the strengths of your study and revise your limitations, they are generally more related with the design.
Response: We have edited the discussion section. And this statement was added: «We cannot exclude the potential confounding effect of some variables on LTFU, which were not collected in this study. We did not investigate whether some missed variables like the body mass index, status of mental illness, stigma and discrimination, distance to the nearest health facility, cell phone possession, and having a caregiver can influence LTFU. »
- Please make sure to discuss what are the implications of your research in actual clinical practice.
This study reports a high LTFU rate in this conflict setting. An ART program in such a setting should pay more attention to naive patients and other particularly vulnerable patients, such as Sudanese patients, during the pre-ART phase. More targeted counseling and follow up is needed. The study implies that the implementation of innovative strategies to address this high risk of being LTFU, reducing either the cost or the distance to the health facility, is warranted. Further studies to analyze whether food security, status of mental illness, stigma and discrimination, distance to the nearest health facility, cell phone possession, and having a caregiver can influence LTFU
We made some edits in the current version.
Minor comments
- Please revise the sub-headings of your table, not clear to me what do you want to say with overall (do you mean all participants, I will advise you to use instead “total”)
Response: Thanks, We made this edits
- Please revise your manuscript and the references for errors.
Response: We have edited
- Please consider using % as numbers instead of “one-fourth”, I will suggest you take your English readable internationally.
Response: The manuscript was edited by MDPI journal service before submission. We will request additional edits from MDPI service
- Please use lower case n, as it denotes the number of people in a sample. An uppercase N represents the number of people in a given population.
Response: Thanks, We made this edits

Reviewer 4 Report
Manuscript Number: ijerph-1591150
Title: Incidence and Predictors of Loss to Follow Up among Patients Living with HIV under Dolutegravir in Bunia, Democratic Republic of Congo: A Prospective Cohort Study
Journal: ijerph
Major comments:
The article is inadequately presented. Furthermore, there are many problems in the different sections as well.
Although the article has scientific rigor, several major flows need to be improved before publication.
1. The title looks unusual.
2. The abstract section is unsuitable—no focus point in the abstract section. Also, so many sentences are fragmented in the abstract section.
3. Rewrite the conclusion (in the abstract) in a more straightforward form. The study implies the implementation of innovative strat- egies to address this high risk of being LTFU, reducing either the cost or the distance to the health facility...…not clear.
4. Authors are suggested to use the full form when used for the first time throughout the manuscript.
5. The current introduction section is concise and contains no new information. Need to write the literature related to anti-retroviral therapy.
6. Delete old references from the introduction section.
7. The introduction section is inapplicable. Need to change the introduction considerably.
8. Aim of the study need to write in a separate paragraph.
9. Need to arrange the introduction section logically.
10. Many grammatically problematic sentences are in different sections, which must be checked and corrected precisely.
11. Study Design and Participants: need to add details.
12. Material and methods are written without proper references. Need a logical flow of the writings with enough references.
13. Teratoma formation, Mouse embryo collection and culture in vitro: need more details.
14. Check all the symbols.
15. Statistical analysis section looks good.
16. The results section need more details.
17. Many grammatically problematic sentences are in the discussion section, which must be checked and corrected precisely.
18. Figure legends are self-explanatory. Need to confirm without the repetition of the results and discussion in the figure legends.
19. The discussion is modest. Please, include the data from other sources about related works.
20. The conclusion needs to address future perspectives.
21. Novelty of the work should be added by the author in the conclusion section.
22. Spacing, punctuation marks, grammar, and spelling errors should be reviewed thoroughly. I found so many typos throughout the manuscript.
23. English is weak. Therefore, the authors need to improve their writing style. In addition, the whole manuscript needs to be checked by native English speakers.
24. References need to increase.
Author Response
Manuscript Number: ijerph-1591150
Title: Incidence and Predictors of Loss to Follow Up among Patients Living with HIV under Dolutegravir in Bunia, Democratic Republic of Congo: A Prospective Cohort Study
Journal: ijerph
Major comments:
The article is inadequately presented. Furthermore, there are many problems in the different sections as well.
Although the article has scientific rigor, several major flows need to be improved before publication.
Response: We have edited where relevant
The title looks unusual.- The abstract section is unsuitable—no focus point in the abstract section. Also, so many sentences are fragmented in the abstract section.
Response: Authors did respect Journal requirement. We have checked and make sure to respect Authors instructions from The journal requirements.
Rewrite the conclusion (in the abstract) in a more straightforward form. The study implies the implementation of innovative strategies to address this high risk of being LTFU, reducing either the cost or the distance to the health facility...…not clear.
A club of patients grouped according to the proximity of addresses, where a patient from the club would be chosen by his or her peers to collect the drugs for the group in order to reduce the risk of insecurity and travel costs when the situation did not allow all the patients to collect the drugs from the clinic
Authors are suggested to use the full form when used for the first time throughout the manuscript.
The current introduction section is concise and contains no new information. Need to write the literature related to anti-retroviral therapy.- The introduction section is inapplicable. Need to change the introduction considerably.
- Need to arrange the introduction section logically.
We did not understand the comments from the reviewers
Delete old references from the introduction section. 8. Aim of the study need to write in a separate paragraph.
Response: We have edited accordingly
Many grammatically problematic sentences are in different sections, which must be checked and corrected precisely.
Response: The manuscript was edited by MDPI journal service before submission. We will request additional edits from MDPI service
Study Design and Participants: need to add details.
Response: We have edited this section.
Material and methods are written without proper references. Need a logical flow of the writings with enough references.
Response: We have edited this section.
Teratoma formation, Mouse embryo collection and culture in vitro: need more details.
Check all the symbols.
Response: although we respect the reviewer view and his comments, we are suspecting that Reviewer is likely examining a different manuscript or mixing manuscripts. We did not report about Teratoma formation, mousse embryo collection and culture in vitro and this make us confused.
Statistical analysis section looks good.
The results section need more details.
Response: We made edit where relevant
Many grammatically problematic sentences are in the discussion section, which must be checked and corrected precisely.
Response: The manuscript was edited by MDPI journal service before submission. We will request additional edits from MDPI service
Figure legends are self-explanatory. Need to confirm without the repetition of the results and discussion in the figure legends.
Response: thanks
The discussion is modest. Please, include the data from other sources about related works.
Response: We have edited
The conclusion needs to address future perspectives.
Response: We have edited accordingly
Novelty of the work should be added by the author in the conclusion section.
Spacing, punctuation marks, grammar, and spelling errors should be reviewed thoroughly. I found so many typos throughout the manuscript.
Response: The manuscript was edited by MDPI journal service before submission. We will request additional edits from MDPI service
English is weak. Therefore, the authors need to improve their writing style. In addition, the whole manuscript needs to be checked by native English speakers.
Response: The manuscript was edited by MDPI journal service before submission. We will request additional edits from MDPI service
References need to increase.
We have edited where relevant
Round 2
Reviewer 1 Report
Given the authors have a finding that there are differences amongst ethnic groups in this paper in terms of follow up - it would be nice to have more background/information and sensitive handling of determinants for why the Sudanese people seem to be more affected.
Not fully confident on the statistical analysis performed - it would be nice in the Tables if they clearly denote which tests were used.
Author Response
Given the authors have a finding that there are differences amongst ethnic groups in this paper in terms of follow up - it would be nice to have more background/information and sensitive handling of determinants for why the Sudanese people seem to be more affected.
Response: The Sudanese seem to be a marginalized and food-insecure people compared to other ethnic groups in the city of Bunia. Previous papers have shown that food insecurity is associated with poor adherence to treatment.
Buju RT, Akilimali PZ, Kamangu EN, Mesia GK, Kayembe JMN, Situakibanza HN. Predictors of Viral Non-Suppression among Patients Living with HIV under Dolutegravir in Bunia, Democratic Republic of Congo: A Prospective Cohort Study. Int J Environ Res Public Health. 2022;19(3):1085. Published 2022 Jan 19. doi:10.3390/ijerph19031085
Not fully confident on the statistical analysis performed - it would be nice in the Tables if they clearly denote which tests were used.
Response: We have added foot notes per request of the reviewer.

Reviewer 4 Report
Manuscript Number: ijerph-1591150
Title: Incidence and Predictors of Loss to Follow Up among Patients Living with HIV under Dolutegravir in Bunia, Democratic Republic of Congo: A Prospective Cohort Study
Journal: ijerph
Major comments:
The modifications are minor. I recommend to modify the manuscript again as per my suggestions.
The article is inadequately presented. Furthermore, there are many problems in the different sections as well.
Although the article has scientific rigor, several major flows need to be improved before publication.
1. The title looks unusual.
2. The abstract section is unsuitable—no focus point in the abstract section. Also, so many sentences are fragmented in the abstract section.
3. Rewrite the conclusion (in the abstract) in a more straightforward form. The study implies the implementation of innovative strat- egies to address this high risk of being LTFU, reducing either the cost or the distance to the health facility...…not clear.
4. Authors are suggested to use the full form when used for the first time throughout the manuscript.
5. The current introduction section is concise and contains no new information. Need to write the literature related to anti-retroviral therapy.
6. Delete old references from the introduction section.
7. The introduction section is inapplicable. Need to change the introduction considerably.
8. Aim of the study need to write in a separate paragraph.
9. Need to arrange the introduction section logically.
10. Many grammatically problematic sentences are in different sections, which must be checked and corrected precisely.
11. Study Design and Participants: need to add details.
12. Material and methods are written without proper references. Need a logical flow of the writings with enough references.
13. Teratoma formation, Mouse embryo collection and culture in vitro: need more details.
14. Check all the symbols.
15. Statistical analysis section looks good.
16. The results section need more details.
17. Many grammatically problematic sentences are in the discussion section, which must be checked and corrected precisely.
18. Figure legends are self-explanatory. Need to confirm without the repetition of the results and discussion in the figure legends.
19. The discussion is modest. Please, include the data from other sources about related works.
20. The conclusion needs to address future perspectives.
21. Novelty of the work should be added by the author in the conclusion section.
22. Spacing, punctuation marks, grammar, and spelling errors should be reviewed thoroughly. I found so many typos throughout the manuscript.
23. English is weak. Therefore, the authors need to improve their writing style. In addition, the whole manuscript needs to be checked by native English speakers.
24. References need to increase.
Author Response
Major comments:
The modifications are minor. I recommend to modify the manuscript again as per my suggestions.
The article is inadequately presented. Furthermore, there are many problems in the different sections as well.
Although the article has scientific rigor, several major flows need to be improved before publication.
The title looks unusual.
The abstract section is unsuitable—no focus point in the abstract section. Also, so many sentences are fragmented in the abstract section.
3. Rewrite the conclusion (in the abstract) in a more straightforward form. The study implies the implementation of innovative strat- egies to address this high risk of being LTFU, reducing either the cost or the distance to the health facility...…not clear.
4. Authors are suggested to use the full form when used for the first time throughout the manuscript.
5. The current introduction section is concise and contains no new information. Need to write the literature related to anti-retroviral therapy.
6. Delete old references from the introduction section.
7. The introduction section is inapplicable. Need to change the introduction considerably.
8. Aim of the study need to write in a separate paragraph.
9. Need to arrange the introduction section logically.
10. Many grammatically problematic sentences are in different sections, which must be checked and corrected precisely.
11. Study Design and Participants: need to add details.
12. Material and methods are written without proper references. Need a logical flow of the writings with enough references.
13. Teratoma formation, Mouse embryo collection and culture in vitro: need more details.
14. Check all the symbols.
15. Statistical analysis section looks good.
16. The results section need more details.
17. Many grammatically problematic sentences are in the discussion section, which must be checked and corrected precisely.
18. Figure legends are self-explanatory. Need to confirm without the repetition of the results and discussion in the figure legends.
19. The discussion is modest. Please, include the data from other sources about related works.
20. The conclusion needs to address future perspectives.
21. Novelty of the work should be added by the author in the conclusion section.
22. Spacing, punctuation marks, grammar, and spelling errors should be reviewed thoroughly. I found so many typos throughout the manuscript.
23. English is weak. Therefore, the authors need to improve their writing style. In addition, the whole manuscript needs to be checked by native English speakers.
24. References need to increase.
Response :
This review , looks like that the reviewer is sharing with us comments from other manuscript not related to the content of our Paper. Indeed, although we are respecting the reviewer views and his comments, we are suspecting that reviewer is likely examining a different manuscript or mixing manuscripts (our manuscript and another manuscript at the same time) or he made copy and paste from his previous review. We did not report about Teratoma formation, mousse embryo collection and culture in vitro and this make us confused. In some cases reviewers embarrassed us by giving feedback that did not allow us to understand the desired changes with generic comments.
